# Susceptibility towards Chickenpox, Measles and Rubella among Healthcare Workers at a Teaching Hospital in Rome

**DOI:** 10.3390/vaccines10101573

**Published:** 2022-09-20

**Authors:** Giuseppe La Torre, Mattia Marte, Valentin Imeshtari, Corrado Colaprico, Eleonora Ricci, David Shaholli, Vanessa India Barletta, Pasquale Serruto, Aurelia Gaeta, Guido Antonelli

**Affiliations:** 1Department of Public Health and Infectious Diseases, Sapienza University of Rome, 00185 Rome, Italy; 2Department of Molecular Medicine, Laboratory of Virology, Sapienza University of Rome, 00185 Rome, Italy

**Keywords:** susceptibility, chickenpox, measles, rubella, healthcare workers, teaching hospital

## Abstract

Immunization is the best protection against chickenpox, measles and rubella. It is important to identify and immunize susceptible healthcare workers to prevent and control hospital infections. Our aim was to estimate the susceptibility level of healthcare workers at a Teaching Hospital in Rome concerning these diseases and the factors associated to the susceptibility. Methods: a cross sectional study was carried out at the Department of Occupational Medicine of the Umberto I General Hospital of Rome. Participants were recruited during routine occupational health surveillance. As far as inclusion criteria, the following professionals were considered: doctors, nurses, laboratory technicians and other health professionals. Concerning the exclusion criteria, patients with HIV, cancer and diseases of the immune system, and acute illness or fever more than 38.5 °C, were not included in the study. A blood sample was tested for the presence of antibodies against measles, rubella and chicken pox. Results: 1106 healthcare professionals were involved in the study (41.8% nurses, 30.4% doctors, 12.3% laboratory technicians, 15.1% other health professionals): 25 (2.3%), 73 (6.6%) and 35 (3.2%) of these were susceptible to measles, rubella and chicken pox, respectively. The only variable associated with susceptibility of measles was age (*p* < 0.001). Furthermore, there was evidence of an association between various susceptibilities, particularly between measles and chickenpox (OR: 4.38). Conclusion: this study showed that even if the majority of our healthcare professionals are immunized for MRV, it is necessary not to underestimate the seronegativity of non-immune ones. All health professionals should be vaccinated to ensure safety for patients, especially the weakest.

## 1. Introduction

Vaccination plays a fundamental role in eliminating the diseases for which it is available [1], including measles, chickenpox, and rubella. In Italy, the vaccination doses foreseen for the three pathogens are two: the first scheduled in the second year of life and the second in the sixth year of life. These are administered in association with the vaccination for mumps (quadrivalent MPRV vaccine) [2].

For populations considered to be at high risks, such as healthcare workers (HCW), vaccination is recommended by the national vaccine prevention plan (PNPV) [3]. In these populations there is the possibility of an additional vaccine booster dose to be carried out in case of lack of evidence of the completion of the vaccination cycle, of seronegativity, or a presumably non-protective antibody titer [4].

Vaccination is important because these diseases can infect not only healthcare workers but also hospitalized patients, fragile by definition [5].

In general, there is a lack of a unified global policy towards vaccinations for the diseases examined, an indication of a low level of attention regarding the issue, even among healthcare workers (HCW) [6], as reported from a Danish study [7]. This occurs even if it is estimated that the risk of acquiring pathologies such as, for example, measles is estimated 2 to 19 times higher in HCW than in the general population [8].

Two surveys conducted in Central and South Italy have shown that healthcare workers (HCWs) have a significantly greater susceptibility to preventable disease vaccines (VPDs) [9,10]. Furthermore, several surveys also show that these are below the vaccination coverage level recommended by the Italian Ministry of Health, in particular for diseases such as measles, chickenpox, rubella, mumps, and pertussis [11,12].

In addition, it must be considered that seronegativity is not exclusively related to non-vaccination, but could also be a consequence of a failure to respond to the primary vaccination course.

A study conducted in Italy on the seroprevalence of anti-measles and anti-rubella IgG among medical students and trainees vaccinated in childhood with two doses of MMR showed seronegativity rates of 15% and 9% respectively, seroconversion after a booster dose of MMR occurred in 74% of cases for measles and 98% for rubella [13].

At the European level, a seronegativity rate for measles in HCWs is estimated to be around 6% [14] while a Catalan study examined seroprevalence for rubella which was 97.2% [15].

The study aims to estimate:(a)The susceptibility level of healthcare workers at a Teaching Hospital in Rome, as a high-risk population, concerning VPD and in particular towards chickenpox, measles, and rubella;(b)The factors associated to the susceptibility.

## 2. Materials and Methods

### 2.1. Study Design

A cross sectional study, according to the Strengthening the Reporting of Observational Studies in Epidemiology (STROBE) statement, was carried out between February 2017 and January 2020 [16].

### 2.2. Setting and Sample

Participants were recruited during routine occupational health surveillance in teaching hospital ‘Policlinico Umberto I’ of Rome. As far as inclusion criteria, the following professionals were considered: doctors, nurses, laboratory technicians and other health professionals. These health care workers work in 15 different integrated care departments (DAI) such as surgical, clinical and health services. Concerning the exclusion criteria, patients with HIV, cancer and diseases of the immune system, and acute illness or fever more than 38.5 °C, were not included in the study. 

During medical examination, social-demographic, clinical and occupational data were collected. After obtaining informed consent from the health care workers, in order to control and prevent nosocomial transmission of measles, rubella and chicken pox, a blood sample was taken by the staff of the Occupational Medicine Unit. This sample was then tested for the presence of antibodies against measles, rubella and chicken pox. 

Antibody titration was performed by the staff of the Microbiology and Viral unit using a semi-quantitative immune-enzymatic test (ELISA method) following the protocol suggested by the producer (Immulite 2000 XP, a continuous random access immunoassay analyzer with a maximum throughput of 200 tests per hour, and specifically Immuno Assay System for IgG anti-Rubella and Bep 2000 Advance System for IgG against measles and chicken pox).

The ranges used for the evaluation of VZV-IgG and anti-measles IgG are:-Negative 0—<0.100-Grey zone 0.100–0.200-Positive > 0.200

The ranges used for the evaluation of anti-Rubella IgG are:-Negative 0.0–0.89-Grey zone 0.9–1.09-Positive > 1.10

Sensibility and specificity of the tests were 97%, 99%, 90.6%, and 98.2%, 97%, 100% for measles, rubella and chickenpox, respectively.

Sample size was calculated using the following parameters:-Population size: 6550 HCWs;-Population proportion of seroprevalence: 80%;-Margin of Error: 3%;-Confidence level: 99%.

Sample size calculations indicated the need to recruit at least 1003 HCWs.

### 2.3. Statistical Analysis

Statistical analysis was performed using mean, standard deviation (SD), median, and minimum and maximum values for quantitative variables. For qualitative variables, frequencies and percentages were computed. Student’s *t*-test or the Mann-Whitney U test was applied for two-group comparisons, and ANOVA and the Kruskal-Wallis test were used for comparisons of more than two groups. The Kolmogorov-Smirnov test was used to verify the normal distribution of quantitative variables. The Pearson’s correlation coefficient was computed to estimate the direct or indirect correlation between variables. 

Differences in susceptibility to exanthematic diseases were assessed using univariate analysis.

Multivariate analysis was conducted using a multiple logistic regression model, considering the following explanatory variables: gender, age, type of department, qualification and typology of company (university or hospital). Susceptibility to infection was considered as a dependent variable.

Results of the logistic regression models were presented as Odds Ratio with 95% confidence interval.

All statistical analyses were performed using SPSS for Windows, release 25.0 (IBM, Armonk, NY, USA). The statistical significance was set at a *p*-value of 0.05.

## 3. Results

A total of 1,106 healthcare professionals were examined including 336 doctors (30.4%), 462 nurses (41.8%), 136 laboratory technicians (12.3%), 167 other health professionals (15.1%). Of these 460 were males (41.6%) and 646 females (58.4%). Age ranged from 25.4 to 70.9 years (mean 54.1 ± 8.8 standard deviation).

History of prior infection was reported by 51.8%, 40.7% and 77.4% for measles, rubella and chickenpox, respectively.

In Figure 1 the prevalence of susceptibility towards measles, rubella and chickenpox are reported by type of wards.

### 3.1. Univariate Analysis

#### 3.1.1. Measles

Table 1 shows the results relating to the univariate analysis of the anti-measles antibody titer. The sera of n. 1102 health professionals. Of these, 25 individuals (2.26%) are susceptible. The univariate analysis shows that the only variable associated with susceptibility is age, with the susceptible having an average age of 47.02 against an average of 54.33 in the non-susceptible (*p* < 0.001).

#### 3.1.2. Rubella

Table 2 shows the results relating to the anti-rubella antibody titer. Sera from 1105 healthcare professionals were analyzed. Of these, 73 individuals (6.6%) were susceptible. The univariate analysis shows that no variable is significantly associated with this type of susceptibility.

#### 3.1.3. Chickenpox

Table 3 shows the results relating to the anti-chickenpox antibody titer. Sera from 1106 healthcare professionals were analyzed. Of these, 35 individuals (3.2%) were susceptible. The univariate analysis shows that no variable is significantly associated with this type of susceptibility.

### 3.2. Multivariate Analysis

#### 3.2.1. Measles

Table 4 shows the results relating to measles susceptibility. The analysis shows that the older the age, the lower the likelihood of being susceptible. In relation to Model 2 (which also contains susceptibility to other diseases and PPD positivity as explanatory variables) it appears that susceptibility to measles is associated with all the others but significantly only with susceptibility to chickenpox (risk increased by almost 5 times).

#### 3.2.2. Rubella

Table 5 shows the results relating to rubella susceptibility. The analysis shows that the only significant factor is the association with PPD positivity: the presence of a positive Mantoux is associated with a decrease in the probability of being susceptible to rubella (OR = 0.64).

#### 3.2.3. Chickenpox

Table 6 shows the results relating to chickenpox susceptibility. As seen above, the multivariate analysis shows that there is a significant association between susceptibility to chickenpox and measles susceptibility (OR = 4.38).

## 4. Discussion

Of the 1106 healthcare workers surveyed, 25 (2.3%) 73 (6.6%) and 35 (3.2%) were susceptible to measles, rubella and chicken pox, respectively. These data are indicative of a better situation than that reported a decade ago by other Italian studies [12,13,17,18], showing that numerous efforts have been made to decrease the susceptible population among health care professionals. However, the susceptibility is somewhat high for rubella (almost 7%) and for all viruses in some wards (General surgery, Emergency, Anesthesia and Critical Areas, Internal Medicine and Geriatric Medical Specialties and Neuroscience/Mental Health) indicating the need for both serological surveillance system and vaccination policy for healthcare workers. In some cases, the susceptibility is much higher than that suggested by WHO in the healthcare system. 

Except for a lower chance of being susceptible to measles with increasing age, no significant differences were found for other socio-demographic or occupational variables, such as gender, use in medical/surgical areas (type department) or the relative structure. Different situation reported in a similar study of Qatar hospital, in which the seropositivity rate of measles in physicians (93.8%) was significantly higher than nurses (83.9%) and allied HCWs (80.5%). As for the age, those aged > 40 years were more seropositive than 30–39-year-old workers (79.0%) and HCWs aged < 29 years (81.5%), with a predominance of varicella seropositive rate in HCWs with longer service [19].

Results of an Italian retrospective cohort study in which the long-term immunogenicity of measles vaccine was evaluated in a sample of two thousand immunized (2 doses of measles-mumps-rubella [MMR] vaccine) students and residents of the University of Bari, reported that in 305 of these (15%) no protective anti-measles IgG were detected and this result was not in line with that of our study and with the evidence of the literature. This proportion was higher among subjects who received vaccination at ≤15 months (20%) than in those who received vaccination at 16–23 months (17%) and at ≥24 months (10%) (*p* < 0.0001), indicating the age over 16 months for a better MMR vaccine efficacy due to a better humoral and cellular immune responses. The Cox analysis suggests that a younger age at the time of the first and second doses of MMR vaccine, an older age at enrollment, and a longer time from the second dose of MMR vaccine to the antibody titer evaluation seem to be risk factors for the persistence of circulating antibodies. However, this study shows that after an MMR vaccine booster dose there was a seroconversion of 74% of seronegative HCWs and the overall seroconversion rate after a second dose (booster) was 93% [13]. 

Furthermore, in our study, there is evidence of an association between various susceptibilities, particularly between measles and chickenpox: the operator who is susceptible to an infectious disease is almost always susceptible to others. This fact involves the possibility of using the serological screening of employees to offer a targeted and free vaccination offer, necessary to stop infectious transmission in the nosocomial setting. Assessing the immune status and additional vaccination requirement is really important to prevent hospital outbreaks as well as occupational infections [19,20]. 

Similar immunity values are reported in a Danish study by von Listow et al. (2019) where it was assessed whether Danish pediatric healthcare workers were protected against selected severe vaccine-preventable diseases (VPD). In particular, protective levels of IgG were found for measles (90.3%), rubella (92.3%). Moreover, they found seropositivity for all three MMR components in 421 (75.9%) HCW, lowest in those younger than 36 years (63.3%). The immunity gaps found primarily in young HCW indicate a need for a screening and vaccination strategy for this group. Considering the poor correlation between self-reported immunity and seropositivity, efforts should be made to check HCW’s immune status in order to identify those who would benefit from vaccination [7].

Several studies conducted in European and non-European countries were carried out to determine the seroprotection against measles, rubella and chickenpox among HCWs, but the strength of our study is the large sample on which to perform the statistical analysis, and there is evidence that training on vaccination and mandatory measures may be needed in order to achieve better coverage [10].

In the occupational setting a screening and vaccination strategy for healthcare workers is needed, especially considering that there is usually a low correlation between self-reported immunity and seropositivity [21]. Some Authors suggest to provide measles-mumps-rubella vaccination without performing serologic testing for new HCWs, and the varicella-zoster vaccination only for those who are negative after serologic testing, using the compulsory worker health examinations as a possible setting [22]. 

The present study has some limitations. First of all, the study design is cross-sectional and a causal relationship between susceptibility and risk factors cannot be established. Moreover, the sample represents one fourth of all HCWs in the Teaching hospital and a selection bias cannot be fully excluded due to the voluntary participation in the survey. Finally, due to practical reasons related to the occupational health surveillance of HCWs, the study was carried out in a three-year period, and this can have introduced some confounding factors.

## 5. Conclusions

In conclusion, even if our study shows that the majority of healthcare professionals are immunized for MRV, it is necessary not to underestimate the levels of seronegativity, that could be the reservoir to the developing of hospital-acquired and occupational infections. All health professionals should be vaccinated to ensure safety for patients, especially the weakest such as children, the elderly and immunosuppressed, remembering that in this segment of the population, a nosocomial infection can lead to the development of serious complications, sometimes even with a fatal outcome. In relation to the results obtained, we therefore suggest implementing the health surveillance program with a careful screening of seroprevalence towards MRV, to promptly recognize susceptible subjects and invite them to vaccination.

## Figures and Tables

**Figure 1 vaccines-10-01573-f001:**
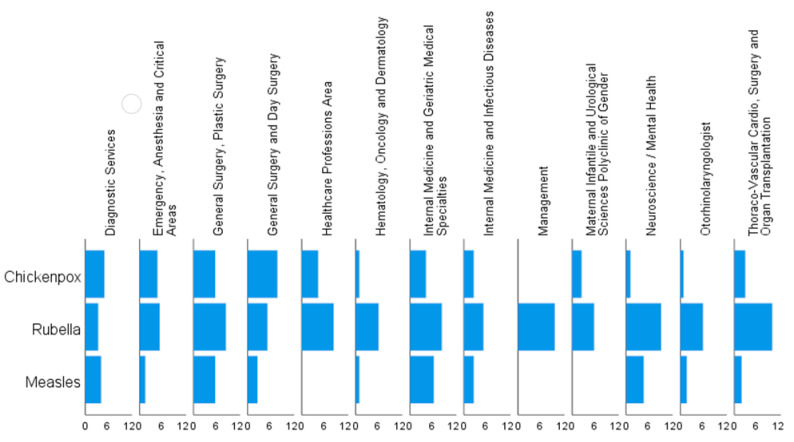
Prevalence of susceptibility towards measles, rubella and chickenpox, by type of wards.

**Table 1 vaccines-10-01573-t001:** Univariate analysis relating to measles susceptibility.

Variable	Susceptible	Not Susceptible	*p*
*Age*	47.02	54.33	<0.001
*Gender*			
Female	15 (2.3%)	626 (97.7%)	0.87
Male	10 (2.2%)	446 (97.8%)	
*Type of Department*			
Healthcare Professions Area	0 (0.0%)	21 (100.0%)	0.175
Thoraco-Vascular Cardio, Surgery and Organ Transplantation	2 (2.0%)	97 (98.0%)	
General Surgery and Day Surgery	1 (2.7%)	36 (97.3%)	
General Surgery, Plastic Surgery	2 (5.9%)	32 (94.1%)	
Management	0 (0.0%)	39 (100.0%)	
Hematology, Oncology and Dermatology	1 (1.0%)	96 (99.0%)	
Emergency, Anesthesia and Critical Areas	3 (1.6%)	179 (98.4%)	
Maternal Infantile and Urological Sciences Polyclinic of Gender	0 (0.0%)	152 (100.0%)	
Internal Medicine and Infectious Diseases	2 (2.7%)	71 (97.3%)	
Internal Medicine and Geriatric Medical Specialties	3 (6.5%)	43 (93.5%)	
Neuroscience/Mental Health	4 (4.8%)	80 (95.2%)	
Diagnostic Services	5 (4.3%)	110 (95.7%)	
Otorhinolaryngologist	2 (1.8%)	110 (98.2%)	
*Role*			
Physicians	8 (2.4%)	329 (97.6%)	0.540
Nurses	10 (2.2%)	448 (97.8%)	
Technicians	5 (3.7%)	130 (96.3%)	
Other Healthcare workers	2 (1.2%)	165 (98.8%)	
*Company*			
Hospital	17 (2.8%)	584 (97.2%)	0.185
University	8 (1.6%)	483 (98.4%)	

**Table 2 vaccines-10-01573-t002:** Univariate analysis relating to rubella susceptibility.

Variable	Susceptible	Not Susceptible	*p*
*Age*	53.45	54.22	0.490
*Gender*			
Female	49(7.6%)	595 (92.4%)	0.120
Male	24(5.3%)	432(94.7%)	
*Type of Department*			
Healthcare Professions Area	2 (8.7%)	21 (91.3%)	0.830
Thoraco-Vascular Cardio, Surgery and Organ Transplantation	10 (10.3%)	87 (89.7%)	
General Surgery and Day Surgery	2 (5.4%)	35 (94.6%)	
General Surgery, Plastic Surgery	3 (8.8%)	31 (91.2%)	
Management	4 (10.0%)	36 (90.0%)	
Hematology, Oncology and Dermatology	6 (6.2%)	91 (93.8%)	
Emergency, Anesthesia and Critical Areas	10 (5.5%)	173 (94.5%)	
Maternal Infantile and Urological Sciences Polyclinic of Gender	9 (6.0%)	141 (94.0%)	
Internal Medicine and Infectious Diseases	4 (5.3%)	71 (94.7%)	
Internal Medicine and Geriatric Medical Specialties	4 (8.7%)	42 (91.3%)	
Neuroscience/Mental Health	8 (9.5%)	76 (90.5%)	
Diagnostic Services	4 (3.5%)	110 (96.5%)	
Otorhinolaryngologist	7 (6.2%)	106 (93.8%)	
*Role*			
Physicians	28 (8.3%)	311 (91.7%)	
Nurses	31 (6.8%)	428 (3.2%)	0.290
Technicians	5 (3.7%)	130 (96.3%)	
Other Healthcare workers	9 (5.4%)	158 (94.6%)	
*Company*			
Hospital	46 (7.6%)	556 (92.4%)	
University	26 (5.3%)	467 (94.7%)	0.113

**Table 3 vaccines-10-01573-t003:** Univariate analysis relating to chickenpox susceptibility.

Variable	Susceptible	Not Susceptible	*p*
*Age*	52.19	54.21	0.200
*Gender*			
Female	24 (3.7%)	620 (96.3%)	0.220
Male	11 (2.4%)	446 (97.6%)	
*Type of Department*			
Healthcare Professions Area	1 (4.5%)	21 (95.5%)	0.830
Thoraco-Vascular Cardio, Surgery and Organ Transplantation	3 (3.0%)	98 (97.0%)	
General Surgery and Day Surgery	3 (8.1%)	34 (91.9%)	
General Surgery, Plastic Surgery	2 (5.9%)	32 (94.1%)	
Management	0 (0.0%)	39 (100.0%)	
Hematology, Oncology and Dermatology	1 (1.0%)	95 (99.0%)	
Emergency, Anesthesia and Critical Areas	9 (4.9%)	173 (95.1%)	
Maternal Infantile and Urological Sciences Polyclinic of Gender	4 (2.6%)	147 (97.4%)	
Internal Medicine and Infectious Diseases	2 (2.7%)	73 (97.3%)	
Internal Medicine and Geriatric Medical Specialties	2 (4.4%)	43 (95.6%)	
Neuroscience/Mental Health	1 (1.2%)	84 (98.8%)	
Diagnostic Services	6 (5.2%)	109 (94.8%)	
Otorhinolaryngologist	1 (0.9%)	112 (99.1%)	
*Role*			
Physicians	10 (3.0%)	326 (97.0%)	
Nurses	13 (2.8%)	449 (97.2%)	0.820
Technicians	5 (3.7%)	131 (96.3%)	
Other Healthcare workers	7 (4.2%)	160 (95.8%)	
*Company*			
Hospital	21 (3.5%)	582 (96.5%)	0.417
University	13 (2.6%)	480 (97.4%)	

**Table 4 vaccines-10-01573-t004:** Logistic regression analysis, dependent variable: measles susceptibility.

Variable	Model 1—Full Model OR (IC95%)	Model 2—With Antibody Titers OR (IC95%)
*Age*	0.93 (0.90–0.97)	0.92 (0.89–0.96)
*Gender*		
Female	1.13 (0.46–2.80)	1.00 (0.39–2.57)
Male	1	1
*Type of department*		
Surgical	0.69 (0.29–1.65)	0.68 (0.27–1.70)
Clinical	1	1
*Role*		
Physicians	1.81 (0.35–9.23)	3.79 (0.44–32.86)
Nurses	1.25 (0.26–5.98)	2.70 (0.33–22.22)
Technicians	3.02 (0.56–16.44)	5.12 (0.54–48.49)
Other Healthcare Workers	1	1
*Company*		
Hospital	0.93 (0.34–2.53)	0.75 (0.27–21.25)
University	1	1
*Susceptibilty*		
Measles		-
Chickenpox		4.80 (1.25–18.42)
Rubella		1.52 (0.31–7.40)
PPD positivity		2.19 (0.72–6.61)

**Table 5 vaccines-10-01573-t005:** Logistic regression analysis, dependent variable: rubella susceptibility.

Variable	Model 1—Full Model OR (IC95%)	Model 2—With Antibody Titers OR (IC95%)
*Age*	1.00 (0.97–1.03)	1.01 (0.98–1.05)
*Gender*		
Female	1.29 (0.74–2.26)	1.31 (0.71–2.42)
Male	1	1
*Type of department*		
Surgical	0.97 (0.58–1.62)	1.12 (0.64–1.95)
Clinical	1	1
*Role*		
Physicians	1.42 (0.63–3.20)	1.24 (0.54–2.89)
Nurses	1.05 (0.48–2.31)	1.02 (0.46–2.38)
Technicians	0.76 (0.24–2.37)	0.58 (0.17–1.97)
Other Healthcare Workers	1	1
*Company*		
Hospital	1.41 (0.78–2.54)	1.33 (0.70–2.52)
University	1	1
*Susceptibilty*		
Measles		1.48 (0.31–7.05)
Chickenpox		2.55 (0.91–7.13)
Rubella		-
PPD positivity		0.64 (0.04–0.69)

**Table 6 vaccines-10-01573-t006:** Logistic regression analysis, dependent variable: chickenpox susceptibility.

Variable	Model 1—Full Model OR (IC95%)	Model 2—With Antibody Titers OR (IC95%)
*Age*	0.97 (0.93–1.02)	0.97 (0.94–1.03)
*Gender*		
Female	1.56 (0.69–3.5)	1.31 (0.71–2.42)
Male	1	1
*Type of department*		
Surgical	1.45 (0.70–2.99)	1.35 (0.65–2.81)
Clinical	1	1
*Role*		
Physicians	0.53 (0.18–1.55)	0.50 (0.17–1.47)
Nurses	0.53 (0.20–1.39)	0.59 (0.22–1.58)
Technicians	0.71 (0.20–2.58)	0.68 (0.18–2.52)
Other Healthcare Workers	1	1
*Company*		
Hospital	0.90 (0.40–2.07)	0.94 (0.41–2.16)
University	1	1
*Susceptibilty*		
Measles		4.38 (1.13–17.03)
Chickenpox		-
Rubella		2.52 (0.91–7.04)
PPD positivity		0.54 (0.15–1.85)

## Data Availability

Data are available upon request.

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
