# Peer review of "Susceptibility towards Chickenpox, Measles and Rubella among Healthcare Workers at a Teaching Hospital in Rome"

_vaccines, 2022, doi:10.3390/vaccines10101573_

Round 1
Reviewer 1 Report
The article titled Susceptibility against chickenpox, measles and rubella among 2 healthcare workers at a teaching Hospital in Rome concluded that majority of our healthcare professionals are immun-25 ized for MRV, it is necessary not to underestimate the seronegativity of non-immune ones. All health 26 professionals should be vaccinated to ensure safety for patients, especially the weakest.
As we all know vaccination provide protection against various infections and diseases. I don't see any novelty in the research findings of the article.
Additionally the discussion and conclusion are not well presented and should be more elaborated with recent references.
Reviewer 2 Report
The manuscript entitled: “Susceptibility against chickenpox, measles and rubella among healthcare workers at a teaching Hospital in Rome ” describes the level of susceptibility level and related risk factors of certain healthcare workers in Rome against chickenpox, measles, and rubella. Its a simple but well-designed cross-sectional study. The analysis is based on robust statistical analysis. However, a major issue missing in this study could be addressed if the following question is answered:
Based on the findings of Cox analysis [stated between lines 183-185] do you recommend any age-specific for having better MMR vaccine efficacy/or minimum age limit for the effective vaccine?
Other comments:
Please explain what is the basis of using that ranges for the evaluation of VZV-IgG and anti-measles IgG in this study.
Most of the findings are presented in tabular form, it will be really very good to present some findings graphically.
Reviewer 3 Report
Susceptibility against chickenpox, measles and rubella among healthcare workers at a teaching Hospital in Rome
The title is misleading, you mean Susceptibility Towards?
Abstract
Immunization is the best protection against chickenpox, measles and rubella. It is important to identify and immunize susceptible healthcare workers to prevent and control hospital infections. Our aim was to estimate the susceptibility level of healthcare workers at a Teaching Hospital in Rome concerning these diseases and the factors associated to the susceptibility. Methods: a cross sectional study was carried out at the Department of Occupational Medicine of the Umberto I General Hospital of Rome. Participants were recruited during routine occupational health surveillance. A blood sample was tested for the presence of antibodies against measles, rubella and chicken pox. Results: 1,106 healthcare professionals were involved in the study (41.8% nurses, 30.4% doctors, 12.3% laboratory technicians, 15.1% other health professionals): 25 (2.3%), 73 (6.6%) and 35 (3.2%) of these were susceptible to measles, rubella and chicken pox, respectively. The only variable associated with susceptibility of measles was age (p <0.001). Furthermore, there was evidence of an association between various susceptibilities, particularly between measles and chickenpox (OR: 4.38). Conclusion: this study showed that even if the majority of our healthcare professionals are immunized for MRV, it is necessary not to underestimate the seronegativity of non-immune ones. All health professionals should be vaccinated to ensure safety for patients, especially the weakest.
COMMENT: to include inclusion and exclusion criteria, sample selection
Major issue
Since the study was an observational study, the approval 228 of the Ethical Committee is not requested. However, the study was carried out following the Hel-229 sinki declaration.
COMMENT: this is inappropriate
METHOD
COMMENT: to include info on
The exclusion criterion, example: was acute illness, fever more than 38.5ºC, recent administration of immunoglobulin, blood products, or immunosuppressive therapy.
-sample size calculation
-This test the sensitivity and specificity ?
-. History of prior chickenpox infection
Study design
A cross sectional study, according to the Strengthening the Reporting of Observa-68 tional Studies in Epidemiology (STROBE) statement, was carried out between February 69 2017 and January 2020 [16].
COMMENT: study span over 3 years? This has many confounding factors
- This sample was 80 then tested for the presence of antibodies against measles, rubella and chicken pox=test methods?
Round 2
Reviewer 1 Report
The authors have revised the manuscript as per the suggestions of the reviewers. However, there are several issues which must be addressed before the publication.
1. Line no. 35 to 36 can be merged/reframed
2. Line no 40 to 43 can be fragmented for the clarity
Check the English and grammar please
3. Line no.43: add comma before or (negativity, or a presumably)
4. Line no. 54: It will be better to provide which country's Ministry. (by the Ministry of Health, which country?).
5. Line no. 281: check the spelling: majorit1y of
Please check the English and grammatical errors throughout the manuscript.
Reviewer 3 Report
Comments are well addressed
Author Response
Please see the attachmen
